

# µID-TIMS: Spatially-resolved high-precision U-Pb zircon geochronology

Sava Markovic[1], Jörn-Frederik Wotzlaw[1], Dawid Szymanowski[1], Joakim Reuteler[2], Peng Zeng[2], Cyril Chelle-Michou[1]

[1]Department of Earth Sciences, ETH Zurich, Zürich, 8092, Switzerland
*Correspondence to*: Sava Markovic (sava.markovic@erdw.ethz.ch)
[2]Scientific Center for Optical and Electron Microscopy (ScopeM, ETH Zürich), Zürich, 8093, Switzerland

**Abstract.** We present a novel methodology for spatially-resolved high-precision U-Pb geochronology of individual growth domains in complex zircon. Our approach utilizes a combined plasma ($Xe^+/Ar^+$) focused ion beam (PFIB)–femtosecond (fs) laser system equipped with a scanning electron microscope (SEM). This system enables micrometer resolution sampling of zircon growth domains with real-time monitoring by cathodoluminescence SEM imaging. Microsamples are then extracted, chemically abraded, dissolved and analyzed by isotope dilution thermal ionization mass spectrometry (ID-TIMS) to obtain

high-precision U-Pb dates. Because of its superior beam precision (~8–20 µm diameter), cleaner cuts, and negligible, nanometer-scale damage imparted on the zircon structure, PFIB machining (30 kV) is preferred for microsamples of sizes expected in most future studies focusing on texturally complex natural zircon (20–120 µm length scales). Femtosecond laser machining is significantly faster and therefore more appropriate for larger microsamples (>120 µm length scales) but it is also coarser (≥20 µm probe size), produces rougher cuts, and creates a minimum of two orders of magnitude wider (micrometer-

scale) structurally damaged zone along the laser cuts. Our experiments show that PFIB machining can be conducted on zircon coated with carbon (minor drift of ion beam during machining) and protective metal coatings (no CL signal) as neither offset the U-Pb systematics nor do they introduce trace amounts of common Pb. We used $Xe^+$ PFIB and femtosecond laser to obtain U–Pb dates for Mud Tank and GZ7 zircon microsamples covering a range of sizes ($40 \times 18 \times 40$ µm – $100 \times 80 \times 70$ µm) and found that microsampling does not bias the accuracy of the resulting µID-TIMS U-Pb dates. The accuracy and precision of

µID-TIMS dates for zircon of any given age and U concentration depend, as for non-microsampled zircon, on $U_{total}/U_{blank}$ and $Pb^*/Pb_c$ – both a function of sample size. Our accompanying open-source code can aid researchers in estimating the necessary microsample size needed to obtain accurate dates at precision sufficient to resolve the processes under study. µID-TIMS bridges the gap between conventional bulk-grain high-precision dating and high-spatial resolution in situ techniques, enabling the study the timescales of a variety of processes recorded on the scale of individual growth zones in zircon. This method

can be applied to zircon of any age and composition, from terrestrial systems to precious samples from other planetary bodies.



## 1 Introduction

High-precision U-Pb zircon geochronology has revolutionized the Earth sciences by providing a numerical calibration to the
geological time scale and quantifying ages and rates of processes from planetary accretion to impacts, supereruptions, and
mass extinctions (Bowring et al., 1998; Bowring and Schmitz, 2003; Schaltegger et al., 2008; Blackburn et al., 2013; Iizuka et
al., 2015; Schoene et al., 2015, 2019; Wotzlaw et al., 2015; Davies et al., 2017). The method relies on accurate and precise
measurements of isotope ratios of two parent-daughter systems of uranium and lead ($^{238}U/^{206}Pb$ and $^{235}U/^{207}Pb$), the proportion
of which is a function of time elapsed since a zircon crystallized. Several features of the zircon U-Pb system have established
it as the most reliable and widely applied geochronometer: 1) zircon is robust under a range of geological conditions, 2) two
independent U-Pb decay systems enable for testing of closed system behavior, 3) negligible amounts of non-radiogenic (i.e.,
common) Pb are incorporated in zircon during crystallization, and 4) zircon is widespread in crustal rocks.

Zircon growth zones record ambient conditions in host magmas either during continuous crystal growth (simple cooling
histories versus episodes of magma reheating or mixing) over timescales of $10^4$–$10^6$ years or during punctuated crystallization
episodes that can be thousands to millions of years apart (e.g., Corfu et al., 2003; Hawkesworth et al., 2004; Costa et al., 2008,
2020; Wotzlaw et al., 2012; Samperton et al., 2015; Chelle-Michou et al., 2017; Szymanowski et al., 2017, 2023; Farina et al.,
2018; Curry et al., 2021; Tavazzani et al., 2023). However, quantifying rates of processes recorded as textural and
compositional complexities in individual zircon crystals is challenging by current analytical techniques applied to U-Pb
geochronology. Isotope dilution thermal ionization mass spectrometry (ID-TIMS), where whole grains or crystal fragments
are dissolved for analysis, grants the required analytical precision but necessarily neglects intra-grain age complexities,
collapsing the entire zircon growth history into a precise, volume-averaged date (Schoene, 2014; Schaltegger et al., 2015;
Schoene and Baxter, 2017). On the other hand, in situ dating with laser ablation inductively coupled plasma mass spectrometry
(LA-ICP-MS) or secondary ion mass spectrometry (SIMS) enables targeting individual growth zones (10–50 µm-wide beam
size) but usually at insufficient precision to resolve intra-grain age differences. An analytical protocol that would combine the
best of both worlds to accurately resolve intra-grain age differences in complex zircon at an age resolution better than the
timescales of investigated processes has thus been a long envisaged but as yet unattained goal of the U-Pb geochronological
community (Grünenfelder, 1963).

Physical sampling of crystal domains within individual zircon for high-precision dating has been conceived as a way to
overcome the respective limitations of in situ and bulk grain dating. Over the last decades, researchers have increasingly
sectioned zircon with mechanical tools such as a scalpel or using a nanosecond laser for ID-TIMS analyses, although such
sampling has been coarse and largely neglected the requirement of textural homogeneity of isolated fragments (e.g., Kovacs
et al., 2020; Samperton et al., 2015). Despite some success, this approach remains arguably approximate and applicable only
to grains with simple internal age relationships. More recently, White et al. (2020) introduced a focused ion beam sampling



technique in petrographic context for U-Pb dating of homogeneous baddeleyite crystals which shows great potential to be adapted for microsampling of zircon.

In pursuit of dating individual growth domains in complex zircon at high precision, we present a methodology that we call µID-TIMS. Our method utilizes a coupled plasma focused ion beam (PFIB) – femtosecond laser machining system for

texturally controlled zircon microsampling in preparation for high-precision dating by CA-ID-TIMS. We first discuss the overall performance and applicability of PFIB and femtosecond laser machining for zircon microsampling. Then, we evaluate the impact of the microsampling procedure (coating and structural damage) on U-Pb zircon systematics. We then present U-Pb isotope results for a number of PFIB- and femtosecond laser-machined microsamples of the Mud Tank (~700–730 Ma) and GZ7 (~530 Ma) zircon reference materials to discuss the accuracy and precision of U-Pb dates obtained with our method.

Finally, we present a code for assessing the feasibility of a zircon microsampling study in terms of accuracy and precision, and discuss future research applications of µID-TIMS.

## 2 Rationale for using plasma focused ion beam (PFIB) and femtosecond laser for zircon microsampling

An adequate machining tool for zircon microsampling should ideally satisfy three criteria: (1) fine enough machining precision (i.e., beam size and sharpness) to ensure microsampling of homogeneous growth zones; (2) microsampling should be

manageable within workable times, as these are priority for conducting a cost- and time-effective study; (3) machining should introduce no bias in the U-Pb systematics of analyzed microsamples.

Figure 1 compares the resolution and speed of different machining techniques. Zircon microsampling requires precision from a few micrometers, for finest cuts, to tens or hundreds of micrometers, for faster machining of larger volumes. Gas field ionization (GFIS) and liquid metal ion source (LMIS) FIB are likely too fine and too slow for the purpose. Plasma focused ion

beam (PFIB) employing different ion species ($Xe^+$, $Ar^+$, $N^+$, $O^+$) covers the range of machining precision required for zircon microsampling from micrometers to tens of micrometers by varying the ion beam current (1 pA to 4 µA) and we found that $Ar^+$ and $Xe^+$ perform best on zircon in terms of milling speed and cut quality. The 515 nm wavelength femtosecond laser has somewhat larger beam size (>20 µm) compared to PFIB operated at highest currents but achieves an order of magnitude faster milling (>10,000 µm³/min compared to <1,000 µm³/min; Fig. 1). Thus, of the currently available micro-machining techniques,

we identify PFIB and femtosecond laser as tools with high potential for the application to zircon microsampling. Both methods are explored below, focusing in particular on associated structural damage, effects of coating, and the quality of the obtained U–Pb data.


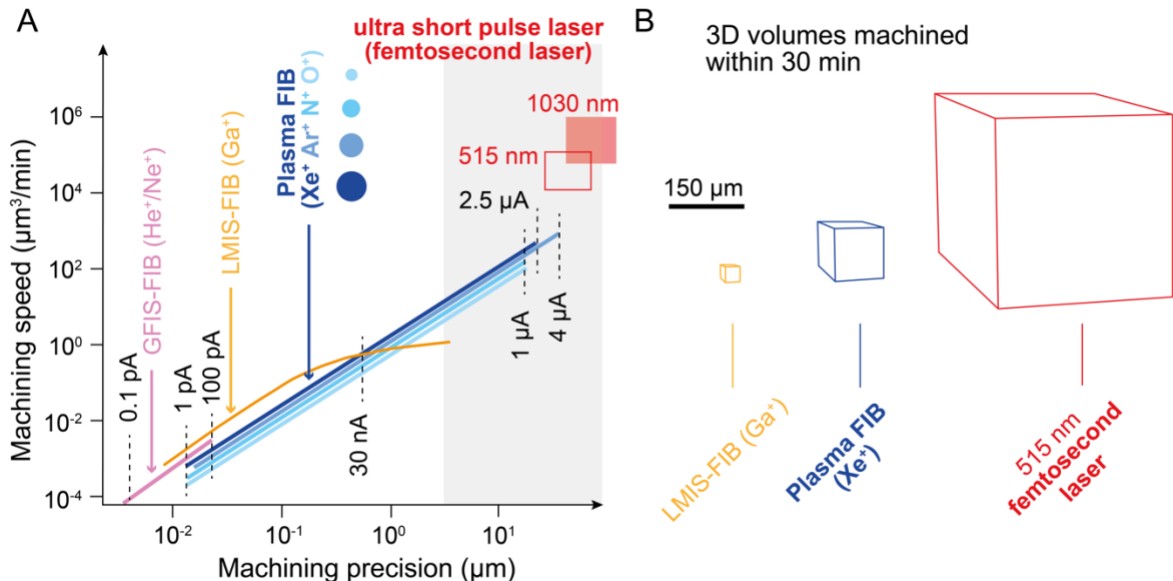

**Figure 1: (A) Precision versus speed of different machining techniques (e.g., Smith et al., 2014; Echlin et al., 2015). Precision (beam size) is shown as a function of ion species for FIB and different wavelengths (515 nm and 1030 nm) for femtosecond laser. (B) Volume**

**of Si machinable within 30 min with the Ga⁺ FIB (100 nA ion beam current) and Xe⁺ PFIB (~2 μA), and femtosecond laser operated at 515 nm wavelength. FIB-focused ion beam, GFIS-gas field ionization source, LMIS-liquid metal ion source.**

## 2 Materials and experimental protocol

### 2.1 Textural and chemical characterization of zircon

Test samples used in this study include megacrysts of the natural zircon reference materials Mud Tank (700–730 Ma and 9
ppm U; Woodhead and Hergt, 2005; Horstwood et al., 2016; Gain et al., 2019) and GZ7 (530 Ma and 650 ppm U; Nasdala
et al., 2018), and zircon crystals from the Lava Creek Tuff (LCT-A; 0.63 Ma and 1500 ppm U; Wotzlaw et al., 2015) and the
Owen Gully Diorite (OG-1; 3.46 Ga and 59 ppm U; see Supplementary Figure S1 and Table S1; Stern et al., 2009; Kemp et
al., 2017). All zircon crystals were first thermally annealed at 900 °C for 48 h, embedded in 1-inch epoxy mounts, and ground
and polished to expose crystal interiors. Cathodoluminescence (CL) imaging of the internal texture of zircon was carried out
on a JEOL JSM-6390 LA scanning electron microscope (SEM) equipped with a Deben Centaurus CL detector or on a Quanta
200F FEG-SEM with a Gatan MiniCL system. In situ trace element composition and U-Pb isotope systematics of the test
crystals were analyzed with a S155-LR ASI Resolution 193 nm excimer laser ablation (LA) system coupled to a Thermo
Scientific Element-XR sector-field inductively coupled plasma mass spectrometer. Trace element and U–Pb isotope signals
were collected simultaneously employing a laser spot-size of 29 µm, a repetition rate of 5 Hz, and an energy density of 2 J cm⁻
². GJ-1 zircon and NIST 610 glass were used as primary reference materials for U-Pb dating and quantifying element
concentrations, respectively, whereas zircons 91500, AUSZ7-1, AUSZ7-5, Plešovice, and Temora served as secondary





reference materials (Wiedenbeck et al., 1995; Wiedenbeck et al., 2004; Black et al., 2004; Jackson et al., 2004; Slama et al., 2008; Kennedy et al., 2014; Von Quadt et al., 2016). Raw output data were processed using Iolite 4 (Paton et al., 2011).

## 2.2 Plasma focused ion beam (PFIB) and femtosecond laser zircon micromachining

Zircon microsamples were machined on a Helios 5 Laser Hydra UX system (Thermo Fisher Scientific), whereas ion irradiation experiments (Section 3.2) were additionally carried out on a Fera3 $Xe^+$ PFIB system (Tescan; ScopeM, ETH Zurich). The Helios 5 system integrates multi-ion-species Plasma FIB ($Xe^+$, $Ar^+$, $O^+$, $N^+$), a femtosecond laser, and an SEM equipped with an external CL detector in one device (Fig. 2). The three columns are fixed in space and zircon machining from different angles is achieved through sample rotation (0–360°) and tilt (-10–57° for standard 1-inch mounts and -38–60° for half-inch mounts and laser objective removed) from the coincidence point of the three columns, or using a pre-tilted sample holder for normal incidence femtosecond laser machining (Fig. 2). Machining and process monitoring are achieved by alternating between seconds- to minutes-long femtosecond laser or PFIB micro-machining steps, and electron imaging on the (CL-)SEM. PFIB machining was done on zircon embedded in epoxy, polished and coated with a 20 nm layer of carbon.

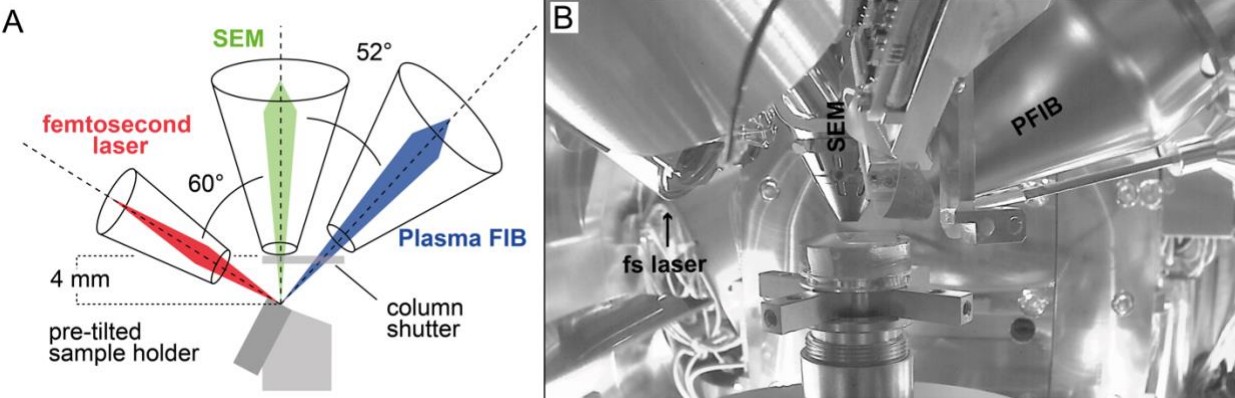

**Figure 2: (A) Geometry of the Helios 5 Hydra PFIB–femtosecond laser-scanning electron microscope system equipped with an external cathodoluminescence detector. (B) Sample chamber view showing the main elements. SEM-scanning electron microscope, GIS-gas injection system.**

PFIB machining (Figs. S2 and S3) involved initial bulk trenching in the front, back and side of the incipient microsample with a normal incident angle ion beam operated at highest current (2.5 µA). Removing enough material in these initial steps is important to provide open space laterally to achieve enough depth (i.e., avoid material redeposition) in the following steps. Back and front trenches were then progressively deepened and microsamples shaped into a trapezoid geometry with a lower current (1 µA) ion beam at a 45 ± 4° incident angle (3° to 11° stage tilt). The microsamples were finally undercut and left



attached to the rest of the zircon only via a narrow side bridge (0.5–1 µA at ~45° beam incident angle). Zircon machining was performed a 30 kV ion beam voltage. The machining depth was controlled by applying an ion dose (in µm, calibrated to silicon milling) three times the desired depth (empirical relationship for zircon milling).

Femtosecond laser machining was performed at a 60° angle with respect to the mount surface (i.e., no use of the pre-tilted holder) employing a 515 nm laser wavelength (Fig. S4). The microsamples were isolated from the surrounding zircon by successive machining and 90° rotation to achieve enough depth, before being undercut. At each step, femtosecond laser was operated at a 6 kHz pulse frequency, 0.048 W power, and pulse energy of 8 µJ, employing vertical and horizontal polarization.

### 2.3 Transmission electron microscopy of PFIB and femtosecond laser irradiated zircon

Structural damage (amorphization, creation of point defects and ion implantation) induced during PFIB and femtosecond laser microsampling was analyzed on a high-resolution (~0.16 nm) Talos F200X transmission electron microscope (TEM) at ScopeM, ETH Zurich. For this purpose, electron transparent (≤100 nm-thin) lamellae oriented perpendicular to the ion- and laser-irradiated surfaces, and one additional lamella from a microsample wall, were prepared using a Helios 5UX $Ga^+$ FIB. $Ga^+$ was the preferred ion species for isolating the effects of previous $Xe^+/Ar^+$ irradiation with high precision. The TEM images

were acquired with a 200 kV electron beam voltage in both standard TEM and scanning (STEM) mode. Besides structural analysis, element distribution maps of TEM lamellae were acquired using a Super-X energy dispersive X-ray spectrometer (EDS) on the Talos TEM.

### 2.4 High-precision (CA-ID-TIMS) U-Pb zircon geochronology

For CA-ID-TIMS, zircon microsamples were detached from the bulk zircon with a scalpel by breaking the bridge. Zircons
were washed in 6N HCl and $HNO_3$ to remove surface impurities, then loaded into 200 µl PFA microcapsules in 1 drop of $HNO_3$, and chemically abraded (CA) in ~50 µl of added 29 M HF at 190–210 °C between 10–14 h to selectively dissolve domains affected by radiation damage. Following CA, zircon aliquots were washed in 6N HCl on a hotplate and in 3.5N $HNO_3$ in an ultrasonic bath. After washing, the zircons were loaded back into their respective microcapsules, spiked with one drop (3–11 mg) of $(^{202}Pb-)^{205}Pb-^{233}U-^{235}U$ ET(2)535 tracer solution (Condon et al., 2015; McLean et al., 2015), and dissolved over
60 h in ~70 µm of 29 M HF at 210 °C in a Parr bomb. After dissolution, the samples were dried down and redissolved in 6N HCl at 180 °C for several hours, dried down, and then redissolved again in 3N HCl. Uranium and lead were separated from matrix elements using an HCl-based single-column ion-exchange chromatography procedure modified from Krogh (1973) and dried down with one drop of 0.02 M $H_3PO_4$. The samples were re-dissolved in a ~1-2 µl drop of Si-gel emitter (Gerstenberger and Haase, 1997) and loaded onto outgassed zone-refined Re filaments.

Isotope ratios of $UO_2$ and Pb were analyzed on a Thermo Triton Plus TIMS instrument in static mode with Faraday cups connected to $10^{13}$ Ω amplifiers or alternatively by peak hopping on a MasCom secondary electron multiplier (Von Quadt et al., 2016; Wotzlaw et al., 2017). Data reduction, date calculation, and uncertainty propagation were carried out using the Tripoli and ET_Redux software (Bowring et al., 2011) with algorithms of McLean et al. (2011). U-Pb isotope ratios and





corresponding dates were calculated relative to the published calibration of the ET tracer solutions (Condon et al., 2015) using

the decay constants of Jaffey et al. (1971), and assuming U blank mass of $0.32 \pm 0.08$ pg (1SD; see Section 3.3.2) and $^{238}U/^{235}U$ of sample and blank of $137.818 \pm 0.045$ (Hiess et al., 2012). All dates are reported with analytical uncertainties at the 95% confidence level.

## 3 Results and Discussion

### 3.1 Performance differences between PFIB and femtosecond laser machining

Our tests confirm that machining footprint and speed are the main performance differences between the PFIB and femtosecond laser. $Xe^+$ and $Ar^+$ PFIB offer clean machining with no surface debris, and produce sharp cuts with machining precision on the order of ~8–20 µm (for commonly used 0.5–2.5 µA ion beam currents). Thanks to these features, PFIB is particularly suited for machining of microsamples of small (20 µm) to moderate (100 µm) dimensions, where the required machining times range between ~45 min and ~3 h. The applicability of PFIB is reduced for larger volume microsamples (>>150 µm), where the

cumulative machining time becomes prohibitively long. For such applications, faster machining with the femtosecond laser (>10,000 µm³/min compared to <1,000 µm³/min for PFIB) is preferred, either for microsampling from start to finish, or at least for initial trenching of large material volumes. In general, with its larger beam size (≥20 µm), rough cuts and more invasive machining footprint, femtosecond laser machining is arguably already too crude for most microsample sizes expected in future studies on natural zircon (≤100 x 100 x 100 µm³). Compared to the nanosecond laser, femtosecond laser does not

produce severe topography and surface debris (White et al., 2021). On the Laser Hydra instrument used here, femtosecond laser machining requires mounting of zircon in smaller radius (~1 cm) mounts on a pre-tilted (54°) holder to achieve full flexibility of machining angles.

### 3.2 Microsampling-induced structural damage in zircon

Damage caused by the $Xe^+/Ar^+$ ion beam (PFIB) and femtosecond laser microsampling was tested in a series of irradiation

experiments on centimeter-sized, carbon-coated Mud Tank crystals (Figs. S5–S7). Each irradiation employed a normal incidence ion beam and targeted zircon zones of homogeneous cathodoluminescence texture to minimize crystal heterogeneity effects. The PFIB irradiation experiments were carried out at voltage conditions ranging from 3 to 30 kV to assess damage both during fine ion polishing (low voltage) and rough machining (high voltage; Figs. S5 and S6). In each experiment, the ion beam current was adjusted to the best fitting discrete current option for consistency between the ion species ($Xe^+$ versus $Ar^+$)

and different instruments (Fera3 versus Hydra). To balance the reduced material removal rate at low voltages (3 and 5 kV), the ion dose was increased such that the exposed region was milled. This ensures that the produced damage is representative, i.e., corresponds to the damage generated on the walls of a microsample prepared by ion milling. PFIB-induced structural damage was analyzed on cross-sections (i.e., TEM lamellae) from the top surface in normal-incidence experiments. Additionally, we analyzed a side-wall of a $Xe^+$ PFIB-machined microsample of Mud Tank zircon to test how representative



our ion irradiation experiments are of the microsampling procedure (Fig. S8). Femtosecond laser irradiation was performed on an embedded zircon with one side exposed (i.e., edge sample) by cutting the mount laterally with a micro saw (Fig. S7). The exposed edge was irradiated with the femtosecond laser beam oriented parallel to the edge, employing a 515 nm wavelength configuration with vertical polarization, a 60 kHz repetition rate, 0.96 W power, and a pulse energy of 16 µJ.

Our experiments show that PFIB machining produces a topmost amorphized damaged zone that hosts a layer of implanted

ions (Figs. 3, 4 and S9–S11). The thickness of this damaged zone, as well as the depth and thickness of the ion-implanted layer increase from <1 nm to a few 10s of nm with increasing ion beam voltage (3 to 30 kV). At the maximum, 30 kV ion beam voltage, which we apply for microsampling, $Xe^+$ irradiation produces a somewhat thinner (~50 nm for Fera3 and ~40 nm for Hydra) damaged layer compared to $Ar^+$ (~60 nm), making $Xe^+$ the preferred ion species for zircon microsampling. Irrespective of the ion species and applied voltage, the damaged zone exhibits a porous, spongy texture (Figs. S7–S9). The transition

towards the underlying undamaged zircon is marked by a change from amorphous to crystalline matter displaying periodic arrangement of atoms in high-resolution TEM images (Figs. 4, S10 and S11). At highest energies (15 kV and 30 kV), the damaged zone exhibits swellings associated with local enrichments in Zr and Si, and depletion in O.

Femtosecond laser machining damages zircon over micrometer length scales that for smaller microsamples correspond to their entire volume (Figs. 3, 4 and S12). The laser-irradiated zircon is porous, and exhibits fractures and globular (melt?) structures

over the entire extent of the TEM lamellae. In the topmost domains of the TEM lamellae immediately exposed to irradiation, Zr is depleted and Si enriched in increasingly porous domains. The globular structures (100–500 nm-wide), as well as most of the irradiated zircon appear amorphous, with crystalline structure detectable only locally in 0.5–1 nm-wide patches.

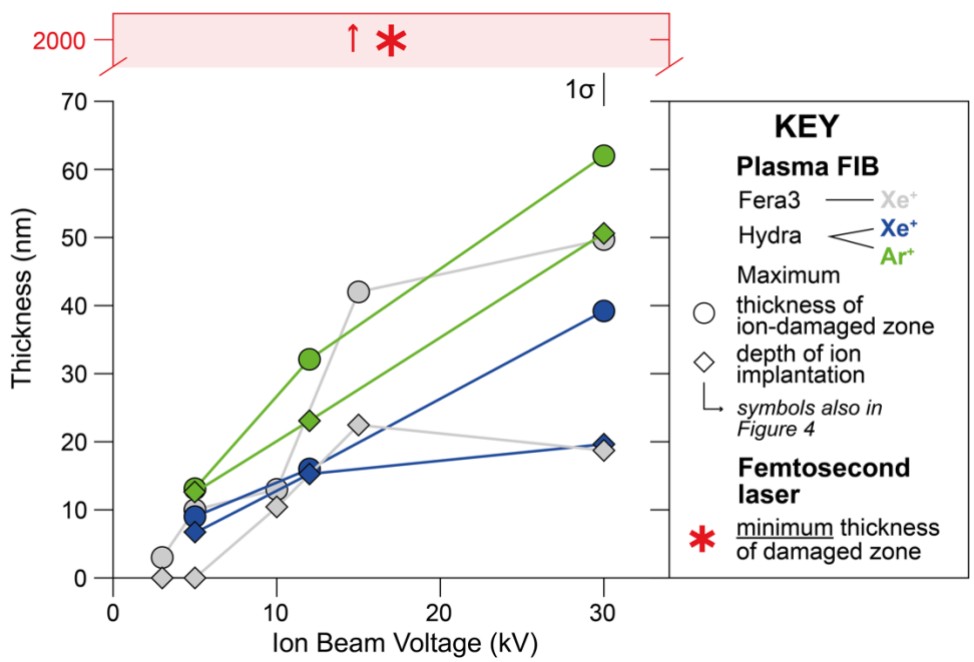




**Figure 3: Structural damage in zircon induced by irradiation with plasma focused ion beam (Xe⁺ and Ar⁺) and femtosecond laser. PFIB damage is quantified in terms of the maximum thickness of the damaged zone and maximum depth of ion (Xe⁺/Ar⁺) implantation. Minimum thickness of the zone damaged by femtosecond laser irradiation is at least two orders of magnitude higher**

**than that generated by PFIB (tens of nm versus >2 μm).**








**Figure 4: (A-H) Transmission electron microscope (TEM) images of the uppermost PFIB-damaged zone. The damaged zircon is**
**porous and amorphous, and shows local swellings. (E-H) EDS maps of the PFIB-damaged zone showing surface enrichment of Zr**
**and Si, and relative depletion of O. (H) Implanted layer of Xe$^+$ within the damaged zone. (I-N) TEM images showing the chaotic**
**texture of zircon damaged by femtosecond laser irradiation. The damaged zone is fractured and contains globular structures (melt?)**
**with only locally preserved crystalline domains (I-L). (M-N) Enrichment in Si and depletion in Zr in the uppermost zone immediately**
**exposed to laser irradiation.**


### 3.3 U-Pb systematics of zircon microsamples

#### 3.3.1 Impacts of Pt-Pd and Cr coating

Thin metal coatings (<20 nm) are commonly used in PFIB machining to reduce beam drift (i.e., deviation of the true compared
to the projected machining pattern), whereas thicker electron or ion beam deposited caps (>20 nm) are typically used to protect
the underlying material from damage caused by the ion beam tails when machining the material adjacent to the cap (Ishitani
and Yaguchi, 1996). Since the amounts of sample Pb and U in dated zircons are small (pg–ng), it is critical that any additional
U or Pb contributions from the coating prior to microsampling can be avoided or corrected for. The impact of Pt-Pd and Cr
protective coatings on U-Pb systematics was therefore tested with high-precision (CA-ID-TIMS) U-Pb isotope analyses of
both metal- and carbon-coated whole zircon crystals undergoing the same preparation steps (Table S2). Carbon-coated crystals
served as a benchmark group. The ~0.63 Ma Lava Creek Tuff Unit A (LCT-A; Wotzlaw et al., 2015) and ~3467 Ma Owens
Gully Diorite (OG-1; Kemp et al., 2017; Stern et al., 2009) zircon were chosen as young, U-rich/Pb-poor and old, U-poor/Pb-
rich end-members, respectively (Figs. S1 and S13). Importantly, for both samples, all individual zircon dates acquired so far
by CA-ID-TIMS using an EARTHTIME tracer solution overlap within uncertainty (Wotzlaw et al., 2015; Laurent et al., 2020),
which allows identifying excess dispersion introduced by the coating. A first random selection of crystals from both zircons
was imaged by CL-SEM to assess within-sample textural variability (Fig. S13). Subsequently, a second random selection of
crystals was hand-picked and separated into three aliquots, with care taken to avoid bias based on crystal size and habit. Each
was mounted in epoxy, and the exposed surfaces of the first two aliquots were coated with a 20 nm layer of Pt-Pd and Cr,
while the third aliquot was coated with 20 nm of carbon.

We find no systematic bias in U-Pb systematics of metal-coated zircon compared to the benchmark zircon coated with carbon
(Fig. 5). For the LCT-A zircon, the $^{230}$Th-corrected $^{206}$Pb/$^{238}$U dates of the Pt-Pd- and Cr-, and carbon-coated zircon all overlap
within uncertainty at ~0.63 Ma, and each group yields weighted mean dates equivalent to those of Wotzlaw et al. (2015; Fig.
5A). Pt-Pd-coated crystals show somewhat higher common Pb masses (Pb$_c$) of 0.39 ± 0.12 pg (mean ± 1SD) compared to 0.12
± 0.05 pg for C-coated crystals, while the Cr-coated crystals (0.27 ± 0.14) are indistinguishable from the two groups (Fig. 5B).



The $^{207}$Pb/$^{206}$Pb dates of the metal- and carbon-coated aliquots of the OG-1 zircon plateau at ~3466.4 Ma, consistent with the
results of Laurent et al. (2020; Fig. 5D). Few younger dates, between 3464–3466 Ma, are recorded for crystals chemically
abraded at 190 °C irrespective of the applied coating, which is suggestive of unmitigated Pb loss. The equivalent common Pb
mass (Pb$_c$) for OG-1 zircon coated with metal (0.27 ± 0.10 pg for Pt-Pd and 0.28 ± 0.10 pg for Cr) and carbon (0.37 ± 0.07
pg), together with overlapping dates for different groups of both LCT-A and OG-1 zircon, indicate that any contribution of Pb
from the metal, if present, was efficiently removed during chemical abrasion both at 190 °C and 210 °C. The different coating
groups are also indistinguishable in terms of total (sample + blank) U mass (Fig. 5F). The metal-coated crystals subjected to
chemical abrasion at 190 °C yield on average higher total U mass (up to 500 pg) compared to the ones abraded at 210 °C (<250
pg), in line with the greater amount of dissolution of high-U zones observed at higher chemical abrasion temperatures
(McKanna et al., 2024). Since no measurable effect of coating material on Pb and U mass was observed, in our further
microsampling work we only applied carbon coating. This allowed us to shorten sample preparation time (i.e., skip repolishing
and coating with metal prior to microsampling) and to make use of CL-SEM imaging in between machining steps. Occasional
beam drift (<<20 µm) was mitigated by re-coating the mount with a new 20 nm layer of C after each ~20 hours of PFIB
machining.








**Figure 5: U-Pb systematics of (A–B) Lava Creek Tuff Unit A (LCT-A) and (C–F) Owen Gully Diorite (OG-1) zircon coated with metal (Pt-Pd and Cr) and carbon. All analyzed LCT-A zircons and one subset of OG-1 zircon were chemically abraded at 190 °C for 14h, whereas another subset of OG-1 zircons was abraded at 210 °C for 10 h. All uncertainties are quoted at 2σ level (95% confidence interval). Pb$_c$-mass of common Pb, U$_{total}$-mass of U from sample and blank.**

**3.3.2 High-precision U-Pb isotope systematics of PFIB and femtosecond laser zircon microsamples**

A total of 29 microsamples of Mud Tank and GZ7 zircon, including those machined with the Xe[+] PFIB and femtosecond laser, were prepared for CA-ID-TIMS U-Pb analysis (Fig. 6 and Table S2). To explore the effect of microsample size (i.e., microsamples' mass of radiogenic lead – Pb* and U) and PFIB/femtosecond laser irradiation on the accuracy and precision of μID-TIMS dates, we machined microsamples covering a range of sizes from $40 \times 18 \times 40$ μm$^3$ to $100 \times 80 \times 70$ μm$^3$. Besides

microsamples, we also analyzed thermally annealed, non-machined, crushed large pieces and shards of Mud Tank and GZ7 zircon as a benchmark group of large and small, non-irradiated aliquots.

Our analyses of the large, crushed pieces of the U-poor (9 ppm) Mud Tank zircon were concordant with $^{206}$Pb/$^{238}$U dates between 708–711 Ma, which we consider a reference for our microsample analyses (Fig. 6E). Note that our analyses yield dates up to 20 Ma younger than the ones published in the literature (Black and Gulson, 1978; Horstwood et al., 2016; Gain et

al., 2019). PFIB microsamples of Mud Tank have larger uncertainties than the crushed pieces, which is explained by their much smaller sizes and resulting lower radiogenic to common Pb ratios (<25 compared to 230–2260). The microsamples and non-irradiated small shards plot along a discordant array extending from a few concordant points between ~700–745 Ma towards a broadly defined zero-age lower intercept (Fig. 6E), consistent with heterogeneous Pb-loss or U-gain.

For the GZ7 zircon (~650 ppm U), our analyses of large pieces are concordant at $529.93 \pm 0.07$ Ma ($^{206}$Pb/$^{238}$U date), providing

a reference date in agreement with the literature (Fig. 6F; Nasdala et al., 2018). PFIB microsamples of GZ7 zircon are also concordant, with some overlapping with the large reference pieces at 529.9 Ma, and others spreading towards younger dates that overlap with the uncertainty of the concordia but describe an array similar to that seen in the Mud Tank data.

From the five microsamples prepared with the femtosecond laser, two largest pieces ($110 \times 105 \times 60$ μm$^3$ and $75 \times 70 \times 45$ μm$^3$) were preserved during chemical abrasion at 190 °C, one disintegrated into shards ($65 \times 45 \times 35$ μm3), while the two

smallest microsamples ($55 \times 55 \times 40$ μm$^3$ and $50 \times 25 \times 30$ μm$^3$) were fully dissolved. We interpret the dissolution of smallest microsamples to be caused by femtosecond laser-induced amorphization that for the smaller volume microsamples may affect their entire volume. The three analyzed femtosecond laser microsamples were also concordant, overlapping partly with the PFIB microsamples, but on average younger than the large pieces at 529.6 Ma.

The observed spread for PFIB and femtosecond laser microsamples compared to reference pieces may either reflect natural

age heterogeneity of the GZ7 crystal on micrometer scale or it is an analytical artifact. The former is unlikely given consistent $^{206}$Pb/$^{238}$U dates of GZ7 at ~530 Ma reproduced by four high-precision laboratories (Nasdala et al., 2018). We identify two

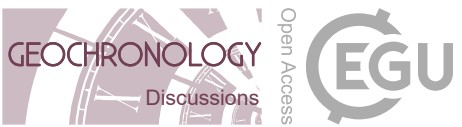

analytical causes that may explain the spread towards younger dates, PFIB- or femtosecond laser-induced Pb-loss or underestimated U blank.








**Figure 6: High-precision U-Pb isotope systematics of PFIB- and femtosecond laser-machined microsamples of Mud Tank and GZ7 zircon. (A) Eleven PFIB microsamples (~40 × 18 × 40 µm³ – 100 × 80 × 70 µm³) within a single crystal of GZ7 zircon. (B) SEM image and simplified sketch (C) of a finalized microsample machined with Xe⁺ PFIB. (D) Top-down view of a femtosecond laser machined microsample. (E-F) U-Pb isotope systematics of PFIB and femtosecond laser microsamples of Mud Tank and GZ7 zircon compared with non-machined, larger pieces and smaller shards.**


We speculate that some Pb loss may occur from the outermost zones of the zircon microsamples damaged during PFIB and femtosecond laser machining (additional heat effect). If not mitigated during chemical abrasion, this would affect the U-Pb isotope data of our microsamples proportionally to the extent of the damaged volume. We computed the PFIB- and femtosecond laser-damaged volume proportion of zircon microsamples as the fraction of the total volume belonging to the

outer damaged zone, excluding the top surface which was not machined. $^{206}Pb/^{238}U$ offset – deviation from accurate dates – for individual microsamples and non-irradiated zircon (i.e., large pieces and shards) was calculated as the relative difference from the average composition of the large pieces for Mud Tank and GZ7 zircon, and the most concordant analyses of LCT-A zircon. For this exercise, we assume no dissolution of the damaged zone during chemical abrasion despite evidence for dissolution of smaller femtosecond laser microsamples. Figure 7 shows a weak correlation between $^{206}Pb/^{238}U$ offset and the

damaged volume for PFIB and no correlation for femtosecond microsamples. For the GZ7 zircon (~650 ppm U), the $^{206}Pb/^{238}U$ and damaged volume of PFIB microsamples show comparable values on the order of <1 %, and similar small offsets values are recorded for femtosecond laser microsamples despite the damaged zone occupying >10 % of their volume. For the U-poor (~9 ppm) Mud Tank zircon, the offset values reach 35% for similar damage volumes, suggesting that the magnitude of offset (degree of inaccuracy) is controlled by a factor specific to each zircon standard. Importantly, these observations imply that

PFIB/femtosecond laser machining prior to ID-TIMS zircon analyses does not introduce bias into U-Pb dates.



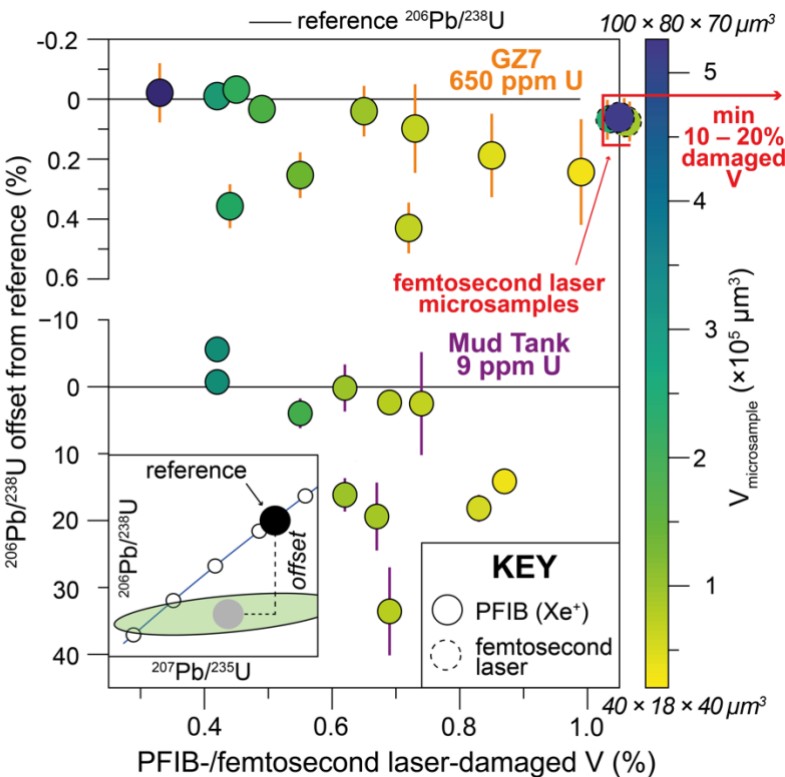

**Figure 7:** $^{206}$Pb/$^{238}$U offset (deviation from accurate dates) versus relative microsample volume damaged by PFIB and femtosecond laser machining. Minimum volume damaged by femtosecond laser is on the order of 10–20 % compared to <1 % for PFIB. Note that the thickness of the damaged zircon zone is constant (39 nm for Xe$^+$ PFIB and minimum of 2 μm for femtosecond laser) and translates into proportionally larger damaged volumes for smaller microsamples. Dimensions (a × b × c in Fig. 6B and C) of the smallest and largest PFIB microsample are given for reference.

Still, within each group (Mud Tank vs GZ7 zircon, PFIB vs femtosecond laser microsamples), the $^{206}$Pb/$^{238}$U offset is correlated with microsample size, with smaller microsamples being more strongly offset (Fig. 7). Extending the analysis to non-microsampled zircon (large pieces, shards and whole crystals), the offset is greatly increased for low U mass (sample + blank) analyses (Fig. 8). Highest offsets (<35%) are reached for analyses with < 160 pg U, whereas lower values mostly below <1% for higher U masses. This observation is consistent with inaccuracies being controlled by isotopic mixing of zircon U isotopic composition with that of the U blank, where low U mass analyses are more affected. Assuming for this purpose that blank U has $^{238}$U/$^{235}$U of 137.818 ± 0.045 representative of magmatic zircon (Hiess et al., 2012), we find that individual Mud Tank microsamples, prepared within a single chemistry, require U$_{blank}$ mass between -0.12 and 1.94 pg to force concordance at our



reference age of 711 Ma (Fig. S14A). Such elevated U blank masses agree with our analyses of total procedural blanks (ET535 and ET2535) between July 2022 and January 2024. From the total of 20 analyses, 16 yield U mass scattering between 0.19–0.49 pg U, with a mean value of 0.32 ± 0.08 pg (1SD), and several outliers reaching ~1 pg and 4.3 pg U (Fig. S14B). This suggests a non-systematic source of U in our analyses which we tentatively link to memory effects of re-used PFA labware. The $U_{blank}$ mass may therefore be an underappreciated source of random and systematic uncertainty, calling for more systematic

monitoring and mitigation, especially prior to analyses of low total U mass zircon (microsamples and whole crystals). Smaller residual offset on the order of <1 %, observed for GZ7 microsamples as well as non-machined zircon arguably reflect heterogeneities in U-Pb isotope composition, minor unmitigated Pb-loss, or still unaccounted for U blank effects. For zircon microsamples, as for non-machined zircon, it follows that for a given laboratory U blank (mass and isotope composition), and zircon of certain age and U concentration, the $U_{total}/U_{blank}$ and therefore U-Pb accuracy becomes chiefly dependent on

microsample size.

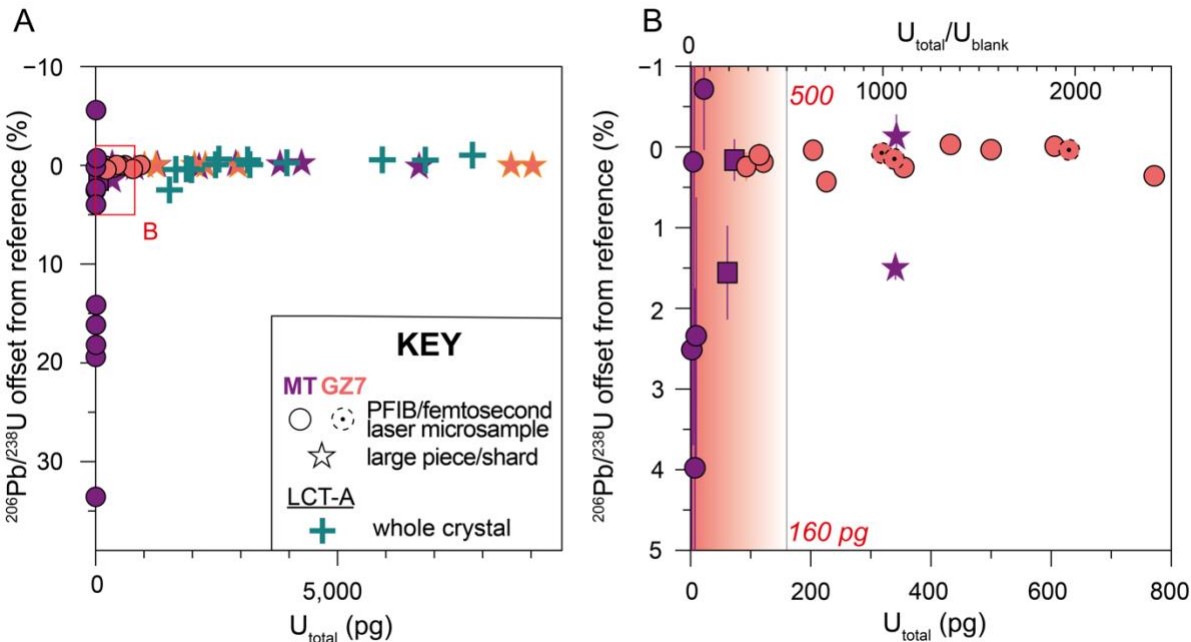

**Figure 8:** $^{206}Pb/^{238}U$ offset as a function of total U mass analyzed (sample + blank) and in relation to U blank mass (fixed at 0.32 ±

**0.08 pg). The offset values increase exponentially to percent levels (<35 %) for both PFIB and femtosecond laser microsamples, and non-machined zircon for $U_{total}/U_{blank}$ below ~500. Residual, <1% offsets observed for $U_{total}/U_{blank}$ are due to heterogeneities in U-Pb isotope composition of dated zircon aliquots, unmitigated Pb-loss, or unresolved U blank effects.**





### 3.3 Designing a µID-TIMS study: minimum zircon microsample size

High-precision U-Pb geochronology of zircon microsamples (µID-TIMS) can be applied to zircon of all ages and U concentrations due to the negligible effects of the microsampling procedures on the U-Pb systematics. As microsample size, U concentration and age determine the total mass of U and radiogenic Pb* for any zircon and these parameters limit the attainable precision of the U-Pb analysis, knowing how small a zircon microsample can be prior to analyses is crucial for planning a successful µID-TIMS study. This is particularly true considering the substantial time and cost factors involved in PFIB microsampling. The ability to precisely and accurately date zircon microsamples also critically depends on the mass and isotopic composition of Pb and U laboratory blank, which should be well characterized in order to plan and execute such a study.

Figures 9 and 10 display the impact of microsample size on the resulting $U_{total}$ and $Pb^*/Pb_c$ for zircon covering a range of U concentrations and age. Given the U blanks measured over the course of this study, an example microsample of $50 \times 50 \times 50$ µm$^3$ dimensions requires a minimum U concentration of ~250 ppm to become relatively insensitive to U blank effects, while low-U (~50 ppm) zircon microsamples require a minimum size of $90 \times 90 \times 90$ µm$^3$ ($U_{total}/U_{blank} > 500$; Fig. 9). Importantly, if the U blank is accurately constrained and corrected for (long-term and within-chemistry $U_{blank}$ mass and isotopic composition), U blank effects do not pose a limit to accuracy of µID-TIMS dates. This is particularly important for $^{206}Pb/^{238}U$ dating of the smallest microsamples (~$20 \times 20 \times 20$ µm$^3$). For old zircons, where $^{207}Pb/^{206}Pb$ dates are quoted, $U_{total}/U_{blank}$ is important to evaluate concordance but the accuracy of $^{207}Pb/^{206}Pb$ dates is unaffected by U blank. For Pb, assuming a constant laboratory Pb blank mass (here 0.1 pg), the minimum microsample size and $Pb^*/Pb_c$ are a function of age and U content. Taking as an example $Pb^*/Pb_c=20$, which represents a minimum ratio where Pb blank correction becomes a minor source of uncertainty (Schoene and Baxter, 2017), relative uncertainty on $^{206}Pb/^{238}U$ on the order of 0.1% can be achieved for moderate to large microsamples (>$80 \times 80 \times 80$ µm$^3$) of older (>500 Ma) zircon richer in uranium (≥500 ppm; Fig. 10). It is noteworthy that regardless of their U concentration, for small microsamples of young zircon (<20 Ma) with low $Pb^*/Pb_c$, Pb blank correction is the main contributor to the total analytical uncertainty on $^{206}Pb/^{238}U$ dates but the absolute precision can still be sufficient to resolve studied processes at these ages.

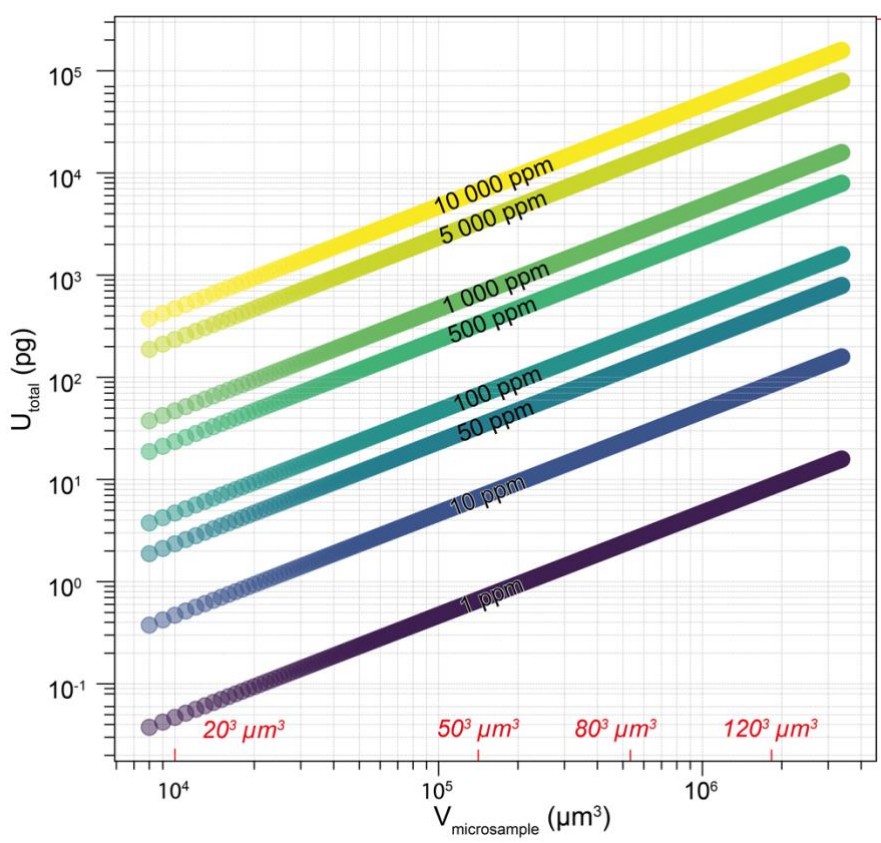


**Figure 9: Microsample total U mass ($U_{total}$) and $U_{total}/U_{blank}$ as a function of microsample volume and zircon U concentration (1–10,000 ppm). Volumes corresponding to cubes of 20, 50, 80 and 120 µm side dimensions are given for reference.**






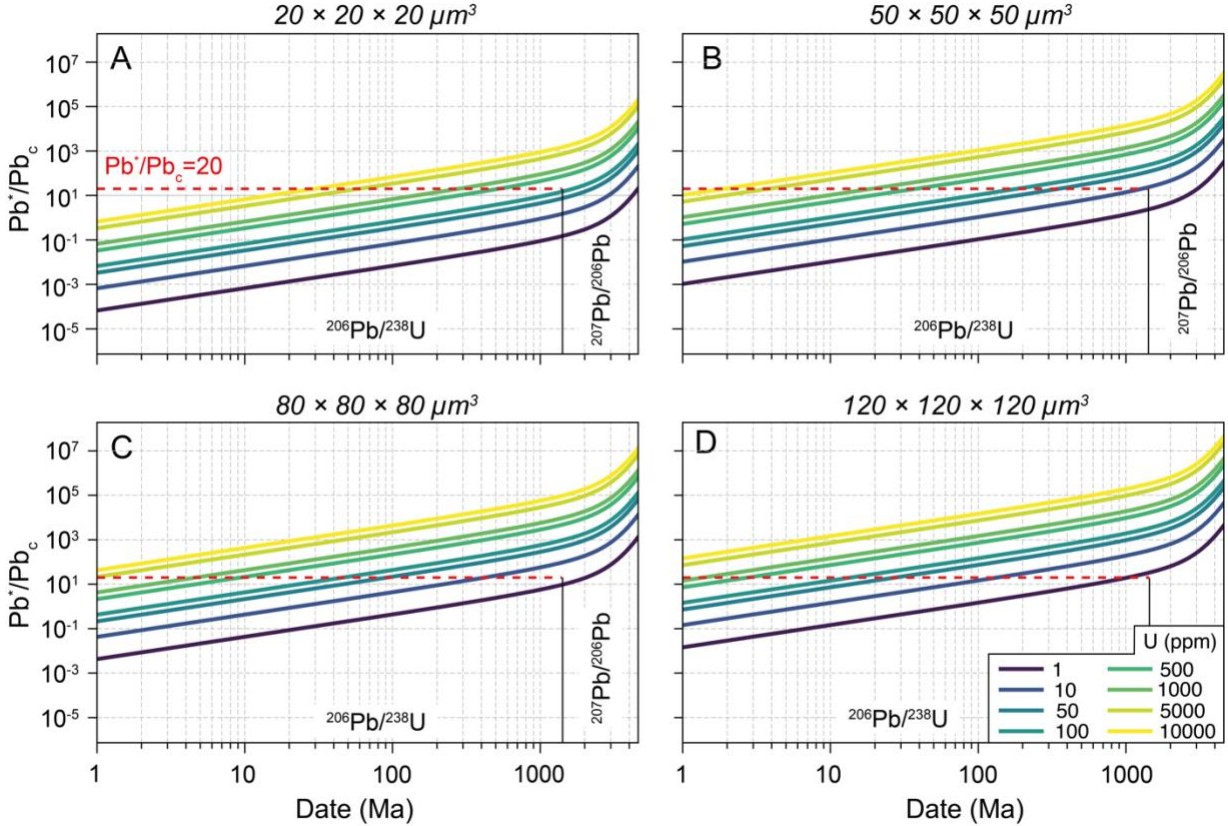


**Figure 10: Theoretical radiogenic to common Pb ratios (Pb\*/Pb_c) microsamples of different size machined from zircon covering a range of ages and U concentration (1–10000 ppm). All models assume a laboratory Pb blank (Pb_c) of 0.1 pg. Pb\*/Pb_c = 20 marks the empirical threshold below which the Pb blank correction dominate the analytical uncertainties on $^{206}$Pb/$^{238}$U dates (Schoene and Baxter, 2017).**


To predict these parameters for each case, we developed an open-source code written in Python that builds on zircon age, composition (estimated or measured U and Th concentrations), and microsample volume, as well as laboratory $U_{blank}$ and $Pb_c$, to compute the analysis' expected $U_{total}/U_{blank}$ and $Pb^*/Pb_c$. These ratios are then evaluated against threshold $Pb^*/Pb_c$ and

$U_{total}/U_{blank}$ in case of $^{206}$Pb/$^{238}$U dates), and, if lower, a new, minimal microsample volume satisfying both conditions is returned. Estimating the precision on a future microsample date is done by comparing the computed $Pb^*/Pb_c$, which is the best predictor of achievable analytical precision, with a compilation of literature CA-ID-TIMS U-Pb and Pb-Pb zircon dates (Fig. 11; Markovic et al., accepted). For different $Pb^*/Pb_c$ (0.1–10 000), the achievable precision on $^{206}$Pb/$^{238}$U ranges from <10 ka



for Cenozoic zircon to between 0.1–1 Ma (including $^{207}$Pb/$^{206}$Pb dates) for older zircon (mostly <0.1–2 %, and <12 % for

Quaternary zircon).












**Figure 11: Achievable 2σ analytical precision on ID-TIMS $^{206}$Pb/$^{238}$U (A-B) and $^{207}$Pb/$^{206}$Pb (C-D) dates of microsamples as a function of zircon age and measured Pb$^*$/Pb$_c$ based on a large compilation of ID-TIMS U-Pb geochronology results (Markovic et al., accepted). For a zircon of given age and U concentration, and known laboratory Pb$_c$ levels (here 0.1 pg), microsample size alone determines Pb$^*$/Pb$_c$. Panel B shows an example of a projected Pb$^*$/Pb$_c$ for a 50 × 50 × 50 μm$^3$ microsample of a 150 Ma zircon with 250 ppm U. In E, best date is defined as $^{206}$Pb/$^{238}$U date for ages ≤1400 Ma, and $^{207}$Pb/$^{206}$Pb for >1400 Ma. Symbols with red outlines are samples analyzed in this study.**


### 3.4 Outlook and potential

μID-TIMS introduces spatially-resolved high-precision U-Pb geochronology. It combines the accuracy and precision
achievable by ID-TIMS with the spatial control of in situ techniques. This methodology can be broadly applied to tackle a
variety of questions related to age determination and timescales not previously accessible to direct quantification. Among
others, they include (1) paired core-and-rim dating to constrain rates of zircon growth across different magmatic environments,
(2) dating of youngest outermost rims of volcanic zircon to more accurately constrain ages of volcanic eruptions, (3) analysis
of young rims in zircon from high-temperature metamorphic terrains to quantify time-scales of crustal melting and zircon
crystallization, (4) extraction of whole zircon or microsamples from thin sections for high-precision geochronology with
petrographic context, (5) dating of precious zircon from meteorites or samples from sample return space missions to investigate
timescales of protoplanetary processes, (6) investigating heterogeneities of U-Pb systematics in zircon reference materials on
the scale of single crystals. Beyond zircon, PFIB and femtosecond laser machining may substitute microdrilling as a more
precise method for obtaining texturally controlled aliquots of complex samples for isotopic analyses, as well as being applied
to microsampling of other U-bearing accessory minerals such as titanite, rutile, apatite and baddeleyite, and to other radiogenic
or stable isotope systems. The microsampling workflow could further be improved through volume imaging of internal zircon
growth zones in absolute space coordinates to fully automate the machining process.

### 4 Conclusions

We present a novel, plasma focused ion beam (PFIB) – femtosecond (fs) laser – CL-SEM machining methodology for
microsampling of zircon fragments for spatially resolved, high-precision U-Pb geochronology (μID-TIMS). Our machining





experiments and tests of the impact of the microsampling methodology on U-Pb systematics of CA-ID-TIMS dated zircon microsamples led to the following findings:

1. PFIB ($Xe^+$/$Ar^+$) is the preferred tool for machining of microsamples of small to moderate dimensions (~20 to 120 µm length scale). Microsamples of these sizes are machined with PFIB within times between ~45 min and 3 h.

2. Femtosecond laser machining is more time- and cost-effective for larger microsamples (>120 µm), especially for applications in which the lower beam precision (≥20 µm), micrometer-scale induced structural damage, and overall larger surface footprint are not a concern – including applications other than zircon U-Pb geochronology.

3. Machining with PFIB can be performed on zircon coated with protective metal (Pt-Pd and Cr) coating or standard carbon coating without introducing any bias to the U-Pb systematics.

4. PFIB and femtosecond laser machining followed by chemical abrasion do not introduce systematic inaccuracies into U-Pb systematics of the analyzed microsamples.

5. PFIB – femtosecond laser microsampling for high-precision U-Pb geochronology can be applied to zircon of any age and U concentration. As for bulk-grain geochronology, the achievable precision is a function of $U_{total}/U_{blank}$ and $Pb^*/Pb_c$, which in turn depend on zircon microsample size, age, U concentration, and laboratory blank. Our open-access code can aid researchers in evaluating these parameters in target zircons, and assess in advance if the achievable analytical precision can resolve the expected timescales of studied processes.

µID-TIMS introduces a new, spatially-resolved high-precision U-Pb zircon dating method, bridging the gap between conventional bulk-grain and in situ dating, opening a number of new applications in Earth and planetary sciences.

*Code and data availability.* Original analytical data and the code presented in this study are available in Supplementary Material.

*Author contributions.* SM collected the data presented in this manuscript, prepared the figures and wrote the manuscript with input from all co-authors. DS, JR and PZ assisted with the data collection and provided input during manuscript writing. Discussion of the data involved all authors. JFW and CCM designed and obtained funding for the study. CCM supervised the study.

*Competing interests.* The authors declare that they have no conflict of interest.

*Acknowledgements.* We acknowledge the The Scientific Center for Optical and Electron Microscopy (ScopeM) of ETH Zurich for providing critical analytical resources for conducting this study. Marcel Guillong and Lorenzo Tavazzani are acknowledged for their assistance with the LA-ICP-MS analyses.

*Financial support.* This research was funded through ETH Zurich Research Grant ETH-05 20-2.





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
