# Peer review of "µID-TIMS: Spatially-resolved high-precision U-Pb zircon geochronology"

_Geochronology, 2024_

## Author Comment (AC1)

Dear Prof. Davis,

Thank you for your insightful comments on our manuscript. Below we address each of the points raised in your review:

If the authors choose not to indent paragraphs you should leave a space between them since otherwise the end of a paragraph is sometimes ambiguous.

We agree that this may not be the clearest formatting choice – however, it was imposed on us by the MS Word template from the *Geochronology* website and the template by default does not include indented paragraphs.

The images shown as Supplementary data files should have captions explaining each of them in detail. There are a lot of complex image figures so it would help the reader if the authors could refer to specific subfigures that illustrate the various features mentioned in the text (e.g. line 221-222). Fig 5 and 6 would be clearer if the sub-figures were labelled with the appropriate sample. If a figure is not broadly comprehensible to the reader without the caption it hasn't been composed well enough. The caption should be there to provide more detail.

In the revised version of the manuscript, we will streamline the presentation of the Supplementary Material to include figures and captions all within in a single pdf file (currently, the captions are provided in a separate pdf file).

Also, we will add the sample labels in Figures 5 and 6.

Do you think it really necessary to perform CA on most zircon samples? CA is very efficient for removing internal altered zones from zircon, which affect only domains that have achieved significant radiation damage. On the other hand, if there is sufficient radiation damage even annealed unaltered zircon will dissolve during the leaching process and Pb can be preferentially leached (a fact that I discovered to my dismay early on and that we tried to explain in Das and Davis 2010 doi:10.1016/j.gca.2010.06.029). I still think that air abrasion is the best method for very old (Archean) zircon. What the authors have developed is similar to air abrasion but much more refined in that one can select internal domains instead of just removing the rims of grains. CA in principle could leach Pb, since it all resides in damaged sites and this was a problem until Mattinson (2005) proposed annealing the sample. In this case, annealing seems to be carried out prior to machining, which means that ion damaged surfaces are exposed to HF during the CA wash. In fact that authors state that the wash is used to dissolve these sites (line 162). However, it is not clear to me whether Pb could be selectively leached compared to U. For example, I would suggest that the distinct discordance shown for Archean standard OG-1 in Fig 5C may have been induced by the CA wash procedure. Any differential leaching will become increasingly evident as the surface to volume ratio increases (i.e. as the sample becomes smaller).

Over the years, chemical abrasion following annealing has been established as the most effective and nearly universally applied procedure for mitigating Pb-loss from radiation damaged zones in zircon (Schoene and Baxter, 2017). While the temperature and duration may be adjusted, we think that it is crucial to maintain this step to ensure the comparability of data obtained from different laboratories (e.g., Widmann et al., 2019, McKanna et al., 2023, 2024). In the case of the machined microsamples, we would expect chemical abrasion to completely dissolve away the outer zircon shell damaged by PFIB or fs laser machining, thereby not fractionating Pb and U. Although we have not studied this directly, total dissolution of the two smallest fs lasered microsamples proves CA effective in removing the laser damaged interface.

Concerning OG-1 zircon, we wish to note that the OG-1 zircon was at no point machined but that the individual analyses are of single whole crystals coated with carbon, Pt-Pd and Cr, all used for testing the effects of different coatings on U-Pb systematics. Since these come in standard crystal sizes and shapes (>80 μm c axis), they have very low surface area to volume ratios and hence do not approach the aspect ratios at which alpha recoil may play a significant role.

In general, discordance in our OG-1 analyses may be the result of one or several factors:

1. Unmitigated Pb loss. At this old age (3.46 Ga), Pb loss is common and one may need to leach at higher temperature to fully remove domains with Pb loss.

2. Cryptic U blank – our individual analyses of OG-1 zircon yield a broad lower intercept at 0, and would require between 0.3 and 0.6 pg of U blank to become concordant. This is overall consistent with the values of our U total procedural blanks measured between July 2022 and January 2024.
3. Preferential loss of Pb through CA (but not from the machined zones since OG-1 was not machined!).

The effect of U blank (2) would have a larger impact for smaller samples. In the Figures below, we show that discordance of OG-1 is most significant for the smallest grains (mass of U and Pb* used as proxies), which suggests that the U blank might be one of the causes for discordance.

[Figure]

On the other hand, the reviewer suggests that scenario 3 should be more important for smaller samples. Here, we don't find a clear size relationship; however, we would like to note the fractionation of $^{207}Pb/^{206}Pb$ evident in the OG-1 samples leached at lower temperatures (see Figure below). It appears that we do see effects produced during (or not removed by) the CA leaching, though they are more evident in $^{207}Pb/^{206}Pb$ than in discordance, and they are not clearly related to size in the same way as discordance above. Whichever the ultimate cause, the result below speaks in favor of leaching at higher temperature as a good solution.

[Figure]

We will briefly mention the different potential causes of discordance in the revised version of the manuscript.

Figs. 6 and 7 show that machined samples can exhibit discordance. This is significant for Mud Tank zircon and much less for GZ7 zircon. This seems surprising since Mud Tank should have much lower radiation damage.

In fact, we believe this relationship to be the key piece of evidence showing that discordance in our dataset is not a geological feature and it is not induced by leaching, but rather a result of U blanks that are unaccounted for. As the reviewer states, Mud Tank should not have any significant radiation damage – and no significant Pb loss. In turn, it does have much lower U contents, so the measured compositions are

extremely sensitive to the applied U blank correction. This is much less the case for GZ7 at 600 ppm U, which despite having significant radiation damage does not exhibit strong discordance.

In the revised version of the manuscript, we will better highlight in the discussion and in Figs. 6 and 7 the key differences between Mud Tank and GZ7 zircon – the extent of radiation damage (Pb loss) and U content.

In addition to the figure that the authors use to illustrate this, it might be interesting to plot discordance versus surface to volume ratio if this can be approximated. As mentioned above, the discordance might have been induced by the CA wash, not necessarily the machining procedure. It might be interesting, for a future study, to try analyzing PFIB machined zircon without applying the HF wash to see if this mitigates the discordance problem. Femtosecond laser machined samples would probably show significant Pb loss from the melt fraction, but this should be removed by the CA leach.

We plot discordance versus surface area to volume ratio for the Mud Tank and GZ7 microsamples prepared with the PFIB (see Figures below). We however note that our current Fig. 7 already includes such a comparison. In our current Fig. 7, offset in $^{206}Pb/^{238}U$ from a non-machined reference zircon is a term similar to discordance whereas relative damaged volume of a microsample is essentially a size term (damaged volume is proportionally larger in smaller microsamples).

In the figures below, discordance correlates positively with the surface area to volume ratio for the microsamples of both Mud Tank and GZ7. It is the much larger absolute values of discordance observed for the U-poor (low radiation damage) Mud Tank (<50 % disc.) compared to the U-richer (higher radiation damage) GZ7 (<2 % disc.) that speak in favor of U blank controlling discordance.

[Figure]

Analyzing a series of PFIB machined zircon microsamples not previously subjected to chemical abrasion would indeed be an interesting test for a future study. The goal of our present study is however to test the effect of microsampling on the U-Pb systematics of samples otherwise prepared the routine way. Testing the impact of chemical abrasion on machined and non-machined samples would be another study with different objectives, and a different design. We nonetheless note that, even if omitting this step were to result in less discordance due to unwanted leaching-induced fractionation of U and Pb, mitigating Pb loss will remain by far the biggest concern in obtaining accurate data.

One significant consideration that should be discussed in the manuscript is the effects of alpha recoil when dealing with tiny samples. The authors should read the excellent Rohmer (2003) paper (DOI 10.1007/s00410-003-0463-0) if they haven't already done so. The problem is that a radiogenic 206Pb atom gets displaced from its parent 238U atom by an average distance on the order of about 50 microns due to the 8 alpha recoil events in the 238U decay chain (Davis and Davis 2018 doi.org/10.1002/9781119227250.ch11) and probably a similar amount for 207Pb. Since this is comparable to machined sample sizes, the Pb that is sampled will not necessarily be representative of the parent U in the sample. At first glance one might think this to be unimportant as long as the sample is distant from a crystal face. However, zircon typically shows micron-scale oscillatory zoning (e.g. LCT-A in Fig S13), where a significant proportion of the Pb in a thin high-U zone will have been displaced outside of it. For example, in my experience, CA treatment of Archean oscillatory zoned zircon resulted in a comb-shaped sample where the high-U zones were dissolved out. The results of ID-TIMS analysis gave data above concordia with a 207Pb/206Pb age slightly too young because of implantation of Pb from the original adjacent high U zones and the difference between average recoil distances for 238U-chain and 235U-chain nuclides. This could be a fundamental limitation on sample size and should be discussed. It might also provide an approach to more accurately measure both 238U and 235U recoil distances.

This is a great point.

On a general note, the final displacement of a daughter isotope ($^{206}$Pb or $^{207}$Pb), even in the statistically rare case of individual displacements along a straight line in one direction, would reach a maximum of only ~0.2 µm (Romer, 2003). These length scales are still significantly smaller than the dimensions of our microsamples, which are mostly >50×40×30 µm. Note that our smallest microsample of ~30×20×20 µm is at the lower limit of sizes that can be manipulated in the lab, and this in itself will prevent one from nearing the volumes at which alpha recoil effects would be stronger.

In our study, we microsampled the internal zones (i.e., away from crystal edges) of Mud Tank and GZ7 zircon, both of which are homogeneous in U content (no sharp contrast between different zones) on the scale of in situ analyses. Chemical abrasion of such "homogeneous" microsamples should therefore not result in fractionation of U and Pb (through selective dissolution of U-richer zones etc.), and this is not of concern for the reference materials used in the present study.

Nevertheless, the effect of alpha recoil may become more significant in future microsampling of strongly zoned natural zircon. Implanting some Pb into the microsample from an adjacent zircon zone that was since removed by machining will affect the microsample rim over an absolute maximum of ~0.2 µm, which is approximately ten times wider than the PFIB damaged zone prior to chemical abrasion. This effect will be clearly more important in microsamples with large surface to volume ratio.

Another interesting scenario is zoning within the microsample which may, in samples that accumulated high radiation damage, result in preferential dissolution of high-U zones, leaving adjacent zones with unsupported excess Pb. However, this is a general concern for application of CA to any sample (and there are examples of reverse discordance likely related to this effect, e.g., in McKanna et al., 2024) and not specific to microsamples. Increasing the temperature of chemical abrasion appears to be helpful in resolving this issue (McKanna et al. 2024), at the obvious risk of losing the sample.

We will add a short discussion on the points from the last two paragraphs, suggesting that these phenomena should be quantitatively addressed in a future study.

Lines 361-365: The author's discussion of the sensitivity of their results to U blank is valid. It might also be due to variability in the assumed isotopic composition of the Pb blank. However, the factors outlined above may provide a more meaningful explanation for discordance. I think that this paragraph should be shortened.

We do not consider the Pb blank composition to be of significance for this issue. The isotopic composition of our laboratory Pb blank is well constrained by many repeated analyses and we fully propagate the uncertainty on that composition into the final U-Pb ratios and corresponding dates. An indirect proof (from the present study) that we correct accurately for the Pb blank is that, for the Pb*-poor LCT-A zircon, we obtain equivalent dates for analyses yielding different masses of Pb blank (Pbc; see Fig. 5).

We will shorten this paragraph as suggested.

Fig 8: I have trouble relating the symbols to the legend. What is the difference between the red symbols with the dot in the middle and those without? Why not just present the symbols in the legend as they are on the figure instead of colour coding the sample names?

At present, the symbols with the dot refer to the microsamples prepared with the fs laser. We understand this might be a confusing way to present data; in the revised version, we will simplify the legend as suggested.

Line 41: 'enable for testing', omit 'for'

We will correct this sentence in the revised version of the manuscript.

Line 143: Should read: 'performed at a 30 Kv ion bean voltage'

We will correct this sentence in the revised version of the manuscript.

Line 219: 'Figs 3, 4(I-M) and S12'

*We will correct this sentence in the revised version of the manuscript.*

Line 221-222: Refer to specific sub-figures that show the features mentioned in the text.

*In the revised version, we will make sure to refer to the specific subfigures.*

Line 253: 'U-Pb isotope analyses were carried out on…' (?)

*We will rephrase the sentence as suggested.*

Line 294: In addition to giving the size range of samples, it might be more meaningful to give their surface to volume ratios.

*We consider that stating the side length of a rectangular prism of volume equivalent to that of microsamples is the most intuitive way of communicating the microsample size.*

*Still, in the revised version of the manuscript, we will also provide the estimated surface area to volume ratios for each microsample in the Supplementary Table. Our microsamples, for which we assume an idealized geometry of a right trapezoidal prism, yield surface area to volume ratios in the range between 0.07–0.20 $\mu m^{-1}$.*

Line 465: I find the authors' reference to volume imaging of zircon to be interesting. I once tried to do this with a UV microscope meant for biological samples but the wavelength was too long to stimulate CL. It might be possible with a short wavelength source and a confocal microscope to produce a tomographic CL image of zircon assuming that UV can penetrate far enough into the grain. Where there is sufficient damage to affect the index of refraction (e.g., Archean zircon), it might be possible to do it optically. I hope that the authors will consider the possibility further.

*Great point – we will explore the possibility of volume imaging further for a future study.*

*On behalf of all the authors,*
*Sava Markovic*

*References:*

1.  McKanna, A. J., Koran, I., Schoene, B., & Ketcham, R. A. (2023). Chemical abrasion: the mechanics of zircon dissolution. *Geochronology*, *5*, 127-151.

2.  McKanna, A. J., Schoene, B., & Szymanowski, D. (2024). Geochronological and geochemical effects of zircon chemical abrasion: insights from single-crystal stepwise dissolution experiments. *Geochronology*, 6, 1-20.

3.  Romer, R. L. (2003). Alpha-recoil in U–Pb geochronology: effective sample size matters. *Contributions to Mineralogy and Petrology*, *145*, 481-491.

4.  Schoene, B., & Baxter, E. F. (2017). Petrochronology and TIMS. Reviews in Mineralogy and Geochemistry, 83(1), 231-260.

5.  Widmann, P., Davies, J. H. F. L., & Schaltegger, U. (2019). Calibrating chemical abrasion: Its effects on zircon crystal structure, chemical composition and UPb age. Chemical Geology, 511, 1-10.

---

## Author Comment (AC2)

Dear reviewer 2,

Thank you for your comments on our manuscript. Below we address each of the points raised in your review:

My first comment is that the authors should more fully recognize the previous work that has been done on laser micro-sampling for CA-TIMS U-Pb zircon geochronology. Specifically, Jim Crowley and Mark Schmitz developed and have been successfully employing laser micro-sampling at Boise State for several years. Unfortunately, they have not published the same type of detailed description of their work; however, examples of their laser micro-sampling can be found in Crowley (GSA abstract, 2018), Kovacs et al. (Engineering Geology, 2020) and Rioux et al. (JMG, 2023). The authors briefly cite the Kovaks study, but group it in with coarse mechanical micro-sampling techniques, and largely dismiss its importance:

"Over the last decades, researchers have increasingly sectioned zircon with mechanical tools such as a scalpel or using a nanosecond laser for ID-TIMS analyses, although such sampling has been coarse and largely neglected the requirement of textural homogeneity of isolated fragments (e.g., Kovacs et al., 2020; Samperton et al., 2015)."

The Boise technique uses thin (~30 microns) doubly polished zircon slabs to fully characterize the zooming patterns and chemistry of the zircon grains, before cutting the grains with a laser. Notably, the spatial resolution of the technique is similar to the current study, with the smallest micros-sampled fractions having weights of 0.1–0.5 micrograms (Crowley, 2018; Kovacs et al. appendix, 2020).

Rather than dismissing this prior work, it would strengthen the manuscript to recognize this alternate micro-sampling technique and discuss the similarities-differences and pros-cons of each method. This prior work in no way takes away from the current study, which is of high quality and very detailed. Laser and PFIB micro-sampling of zircon is a new field and it is highly beneficial to have multiple labs experimenting with different methods.

Thank you for bringing this omission to our attention. We were not aware of the GSA abstract and of the very interesting study of Rioux et al. This is certainly an interesting approach, and indeed it looks like this technique is comparable in spatial resolution to ours. We encourage the authors to document the effect of using the ns laser and the associated damage on U-Pb results.

In the revised version of the manuscript, we will cite the additional work you mentioned and recognize more fully the work on zircon cutting with the ns laser performed at Boise State. Also, we will include a sentence on the comparison of the two approaches.

Secondly, I have some general comments and questions on the origin of the scatter in the micro-sampled zircon dates. The authors did a nice job critically testing whether beam damage impacted the U-Pb systematics and working to understand the scatter in the data. The authors' conclusion that the scatter in the data is not directly related to beam damage seems reasonable, and is expected given that the damaged lattice is likely removed during the chemical abrasion step. As a clear test of this, I liked that the authors analyzed both large and small shards of mechanically broken zircon (i.e. not PFIB or laser cut). I would like to see a larger dataset of analyses of very small mechanically broken shards for both samples, which have volumes similar to the PFIB and laser micro-sampled zircon fragments. Such a data set would allow for a direct comparison between mechanically sampled grains and PFIB and laser micro-sampled grains. If the two datasets do show similar dispersion, it would more definitively rule out beam related lattice damage/Pb loss.

We fully agree with your observations. To gain more clarity on the cause of dispersion, for the revised version of the manuscript, we will analyze a few small pieces (volume similar to microsamples) of both Mud Tank and GZ7 zircon to complement our dataset.

The authors argue that the observed scatter in the micro-sampled fragments is most likely related to variable U blanks, which seems very possible. My most significant concern with this conclusion is that the percentage of impacted grains seems to be inconsistent with the U blank data reported in Fig. S14. The figure shows that over a 1.5 year period, 16 out of 20 U blanks had values of ~0.3 pg; however, if all of the scatter in the micro-sampled zircon dates is due to variable U blanks, Figure S14A suggests it would require U blanks >0.5 pg for 9 out of 13 analyses. The authors should discuss why the U blanks might be more variable in the actual analyses than in their blank measurements. It might also be informative to run an entire batch of

blanks to better understand the variability of the U blanks within a single batch. If there is significant variability, it would suggest that the authors should be using a higher U blank value with larger uncertainties.

Admittedly, we did not anticipate the U blank could cause any issue when we designed the study and until relatively late in the project. Therefore, most of the blanks reported here were measured over a different period (June 2021–January 2024) compared to our samples and might not be directly comparable. Therefore, for the completeness of the study we will also provide more U blank analyses in the revised manuscript.

I am also surprised that the U blanks are so high. U blanks are typically an order of magnitude smaller than Pb blanks, as would be expected, given the much high concentrations of Pb in the environment. The U blank data presented in Figure S14B suggest that the U blanks during the time of this study are similar to or higher than the Pb blanks. This is an unexpected result, and as the authors note, raises concerns about memory effects in the micro-capsules.

We were also surprised to find that U blanks could be as high. Since we made this discovery, we started tracking the U blank more systematically (loading, total procedural and sample beaker blanks).

In the revised version of the manuscript, we will additionally highlight that constraining the U blank and its variability should become standard practice prior to any analyses of low U zircon (especially microsamples).

Even based on the current data and text, the lab appears to be applying a U blank of 0.07 ± 0.02 pg (Fig. S14A), whereas measured U blanks have an average of 0.32 ± 0.08 pg (Fig. S14B).

In the present manuscript, we are applying a U blank mass of 0.32 ± 0.08 pg which is our best estimate of the blank applicable over the study period. The U blank of 0.07 ± 0.02 pg stated in Fig. S14A was the value used prior to our more recent measurements. We will remove this from Fig. S14A for the revised version of the manuscript to avoid confusion.

Detailed comments:

Lines 298–300: "Note that our analyses yield dates up to 20 Ma younger than the ones published in the literature (Black and Gulson, 1978; Horstwood et al., 2016; Gain et al., 2019)."

The authors should discuss this in more detail. Why are their dates 20 Ma younger?

The discrepancy between our dates and the ones from the literature was surprising for us as well. However, we note that our high-precision dataset of large crushed pieces (>1 ng U) is internally consistent, and even with very elevated U blanks, one would not reproduce the ages of Horstwood et al.

One possible explanation for the observed age discrepancies may be natural isotopic variability of Mud Tank zircon, between different crystal and possibly within a single crystal – similar to what Schaltegger et al. (2015) observed for zircon megacrysts from the Alps. We will highlight this point in the revised version of the manuscript.

Lines 462–465: "Beyond zircon, PFIB and femtosecond laser machining may substitute microdrilling as a more precise method for obtaining texturally controlled aliquots of complex samples for isotopic analyses, as well as being applied to microsampling of other U-bearing accessory minerals such as titanite, rutile, apatite and baddeleyite, and to other radiogenic or stable isotope systems."

I recommend changing "substitute" to "replace".

We will change "substitute" to "replace" in the revised version of the manuscript.

Figure 6: It would be useful to add labels to the concordia figure indicating which graph is Mudtank versus GZ-7.

We will add sample labels to the concordia figures (Fig. 6).

On behalf of all the authors,
Sava Markovic

References:

1. Schaltegger, U., Ulianov, A., Müntener, O., Ovtcharova, M., Peytcheva, I., Vonlanthen, P., ... & Girlanda, F. (2015). Megacrystic zircon with planar fractures in miaskite-type nepheline pegmatites formed at high pressures in the lower crust (Ivrea Zone, southern Alps, Switzerland). *American Mineralogist*, *100*(1), 83-94.